# Electron Scattering from 1-Methyl-5-Nitroimidazole: Cross-Sections for Modeling Electron Transport through Potential Radiosensitizers

**DOI:** 10.3390/ijms241512182

**Published:** 2023-07-29

**Authors:** Ana I. Lozano, Lidia Álvarez, Adrián García-Abenza, Carlos Guerra, Fábris Kossoski, Jaime Rosado, Francisco Blanco, Juan Carlos Oller, Mahmudul Hasan, Martin Centurion, Thorsten Weber, Daniel S. Slaughter, Deepthy M. Mootheril, Alexander Dorn, Sarvesh Kumar, Paulo Limão-Vieira, Rafael Colmenares, Gustavo García

**Affiliations:** 1Instituto de Física Fundamental, Consejo Superior de Investigaciones Científicas-CSIC, Serrano 113-bis, 28006 Madrid, Spain or ai.lozano@fct.unl.pt (A.I.L.); lid.alvarez@iff.csic.es (L.Á.); adrian.garcia.abenza@csic.es (A.G.-A.); carlosguerra@iff.csic.es (C.G.); 2Laboratório de Colisões Atómicas e Moleculares, CEFITEC, Departamento de Física, Universidade NOVA de Lisboa, 2829-516 Caparica, Portugalplimaovieira@fct.unl.pt (P.L.-V.); 3Laboratoire de Chimie et Physique Quantiques (UMR 5626), Université de Toulouse, CNRS, UPS, 31062 Toulouse, France; fkossoski@irsamc.ups-tlse.fr; 4Departamento de Estructura de la Materia, Física Térmica y Electrónica e IPARCOS, Universidad Complutense de Madrid, Avenida Complutense, 28040 Madrid, Spain; jaime_ros@fis.ucm.es (J.R.); pacobr@fis.ucm.es (F.B.); 5Centro de Investigaciones Energéticas Medioambientales y Tecnológicas (CIEMAT), Avenida Complutense 22, 28040 Madrid, Spain; jc.oller@ciemat.es; 6Chemical Sciences Division, Lawrence Berkeley National Laboratory, Berkeley, CA 94720, USA; mhasan2@lbl.gov (M.H.); tweber@lbl.gov (T.W.); dsslaughter@lbl.gov (D.S.S.); 7Department of Physics and Astronomy, University of Nebraska-Lincoln, Lincoln, NE 68588, USA; martin.centurion@unl.edu; 8Max Planck Institute for Nuclear Physics, 69117 Heidelberg, Germany; deepthy.mootheril@mpi-hd.mpg.de (D.M.M.);; 9Servicio de Radiofísica, IRYCIS-Hospital Universitario Ramón y Cajal, Carretera de Colmenar Viejo Km. 9.100, 28034 Madrid, Spain; 10Centre for Medical Radiation Physics, University of Wollongong, Wollongong, NSW 2522, Australia

**Keywords:** electron scattering cross-sections, electron impact molecular fragmentation, molecular radiosensitizers, radiation damage, ionization, dissociation

## Abstract

In this study, we present a complete set of electron scattering cross-sections from 1-Methyl-5-Nitroimidazole (1M5NI) molecules for impact energies ranging from 0.1 to 1000 eV. This information is relevant to evaluate the potential role of 1M5NI as a molecular radiosensitizers. The total electron scattering cross-sections (TCS) that we previously measured with a magnetically confined electron transmission apparatus were considered as the reference values for the present analysis. Elastic scattering cross-sections were calculated by means of two different schemes: The Schwinger multichannel (SMC) method for the lower energies (below 15 eV) and the independent atom model-based screening-corrected additivity rule with interferences (IAM-SCARI) for higher energies (above 15 eV). The latter was also applied to calculate the total ionization cross-sections, which were complemented with experimental values of the induced cationic fragmentation by electron impact. Double differential ionization cross-sections were measured with a reaction microscope multi-particle coincidence spectrometer. Using a momentum imaging spectrometer, direct measurements of the anion fragment yields and kinetic energies by the dissociative electron attachment are also presented. Cross-sections for the other inelastic channels were derived with a self-consistent procedure by sampling their values at a given energy to ensure that the sum of the cross-sections of all the scattering processes available at that energy coincides with the corresponding TCS. This cross-section data set is ready to be used for modelling electron-induced radiation damage at the molecular level to biologically relevant media containing 1M5NI as a potential radiosensitizer. Nonetheless, a proper evaluation of its radiosensitizing effects would require further radiobiological experiments.

## 1. Introduction

Using high-atomic number (Z) elements to enhance the energy deposition (absorbed dose) during X-ray irradiation of living tissues was introduced many years ago [1]. Metal atoms (Au, Pt, Gd) and metallic nanoparticles [2,3,4] have been extensively studied as potential radiosensitizers in conventional (photon beam-based) radiotherapy treatments [5]. The absorption coefficient of photons in matter increases with Z, together with the probability of generating Auger electrons [6]. These metallic structures have also been proven to enhance the energy deposition of charged particle beams (electrons, protons and heavy ions) as those used in new advanced radiotherapy techniques, such as electron-flash intraoperative radiotherapy [7], proton therapy [8] and heavy ion (He, C, O) beam radiotherapy [9,10,11]. In these cases, the radiosensitizing mechanisms are not well understood and could be related to induced secondary processes on the surface of the nanoparticles and their molecular coating [12]. Recently, the concept of a molecular radiosensitizer [13] has been introduced as a more appropriate targeting procedure for charged particle beams. These molecules enhance radiation effects in tumoral areas with less toxicity than heavy atom nanoparticles. Essentially, low-energy secondary electrons generated by the primary beam dissociate the molecular radiosensitizer [14] and create abundant reactive radical species that are able to efficiently attach to the DNA molecular components or reduce the tumor hypoxia, which is a barrier to effective radiation therapy [15]. In particular, 1M5NI is a methylated nitroimidazole whose structural schematic is shown in Figure 1. Its radiosensitizing properties, as well as for other imidazole derivatives, are linked to the formation of negative radical species, such as CN^−^, OH^−^, NO_2_^−^ and other reactive anionic species [16]. These radicals are produced by dissociative electron attachment (DEA) processes when low-energy electrons (typically with kinetic energies below 10 eV) are captured by the target molecule and they are very efficiently damaging the DNA. Although these processes are triggered by low-energy electrons, photoelectrons and secondary electrons are generated within a broad kinetic energy range. Note that the main goal of radiotherapy is to damage the DNA of tumoral cells but to preserve, as much as possible, the DNA in healthy tissue. In this context, electron-induced dissociation to potential molecular radiosensitizers are critical processes and it is important to understand their radiosensitizing mechanisms. In order to model the radiation effects when molecular radiosensitizers are deposited in the tumor, the corresponding electron-induced chemistry needs to be characterized in terms of the interaction probabilities (cross-sections) of all the available reaction channels at a given electron energy. These considerations motivated the present research study.

The main goal of this study is to provide a self-consistent electron scattering data set for a potential molecular radiosensitizer, 1-methyl-5-nitroimidazole (1M5NI), within the impact energy range (0.1–1000 eV). 

This consistent data set has been obtained by combining the theoretical and experimental data available in the literature with our own measurements and calculations of the total electron scattering and integral elastic and inelastic cross-sections, as well as the induced cationic and anionic fragmentation.

The remainder of this article is organized as follows: Section 2 provides the results and their interpretation. These results are discussed in Section 3 together with a critical comparison with previous data. The theoretical and experimental methods used in this study are described in Section 4. 

## 2. Results 

In order to obtain a comprehensive self-consistent data set for modelling purposes, we followed the procedure described in previous studies (see ref. [17] and references therein). Essentially, accurate measurements of the total electron scattering cross-sections (TCS) were used as reference values to ensure the self-consistency of the adopted database. The sum of the cross-sections of all the scattering processes, which are available at a given energy (open channels), should give the reference TCS value at that impact energy. In this case, the reference values were derived by combining the present TCS measurements in the range of 0–300 eV with our calculated values for impact energies above 300 eV, as described in the next subsection.

### 2.1. Total Electron Scattering Cross-Sections

As previously mentioned, our recently published TCS measurements [18] were used as the reference values for electron impact energies from 1 to 300 eV. These results were obtained with a magnetically confined electron beam apparatus [19], and their assigned uncertainties are within 5%. The TCS measurements based on the attenuation of a linear electron beam passing through a low-pressure gas cell containing the target of interest are likely the most accurate results that we can use for this purpose. They are affected by the “missing angles” systematic error (see ref. [19] for details), but they have been corrected according to the procedure described below.

For energies above 20 eV, we used our screening-corrected additivity rule within the framework of the independent atom model (IAM-SCAR) procedure [20], including the interference effects (IAM-SCARI) [21] to calculate the differential elastic, as well as the integral elastic, inelastic and, therefore, total scattering cross-sections. This method has proven to be reliable within 10% for a large number of molecular targets [17] and impact energies above 20 eV. In the case of 1M5NI, the agreement of the calculated TCS with the experimental results is excellent [18] for impact energies higher than 20 eV. Accordingly, we used the calculated values to extrapolate the present experimental TCS values up to 1000 eV. 

As described in previous publications [19,22], the calculated differential elastic cross-sections (DCS) can be used to quantify the magnitude of the “missing angle” effect by integrating these DCS values over the acceptance angle of the detector for each electron incident energy. The corresponding results are shown in Table 1. Considering that the experimental values have been corrected for this systematic error, we estimated an overall uncertainty limit of about 10% for the present TCS reference values.

### 2.2. Differential and Integral Elastic Cross-Sections

For energies below 15 eV, we used the Schwinger multichannel (SMC) method [23,24] to calculate the differential and integral elastic scattering cross-sections [18]. 

For intermediate and high energies (15–1000 eV), we used our IAM-SCARI method [20,21]. It is based on an independent atom representation and it includes the screening of the atomic cross-sections within the molecule and considers interference effects due to the multicenter scattering process. Details on the calculation are given in Section 4 (Materials and Methods).

The representative differential elastic cross-sections (DCS) for impact energies ranging from 1 to 1000 eV are shown in Figure 2. The numerical values of these DCS and the corresponding integral elastic scattering cross-sections (ICS), derived by integrating the differential values over the whole scattering angle range (0–180 deg), are shown in Appendix A (see Appendix A). These data were calculated via the procedures above; from 0.1 to 15 eV, the SMC data were chosen, while the IAM-SCARI results were utilized for energies above 15 eV. For the overlapping impact energy (15 eV), there is a good agreement between both methods. At that energy, the SMC results tend to be lower than the IAM-SCARI in the forward direction (close to 0 degrees), which is due to the limited number of partial waves included in this calculation (no dipole Born corrections [25] are included). There is an excellent agreement from 4 to 27 deg, but for the higher scattering angles, although they have a similar shape, the minimum cross-section around 120 deg is much more pronounced in the IAM-SCAR calculation (see Figure 2). Nonetheless, by integrating the whole scattering angle range (0–180 degrees), the agreement between both ICS values (58.64 and 60.48, respectively) was found to be better than 3%. 

### 2.3. Differential and Integral Inelastic Cross-Sections

For energies above the threshold energy to excite the different inelastic channels (rotational excitation, electron attachment, vibrational excitation, electronic excitation and ionization), the differential and integral cross-sections related to the channel in question need to be known in order to obtain a realistic representation of the scattering problem. In the following subsubsections, these inelastic channels will be analyzed.

#### 2.3.1. Electron Attachment Cross-Sections

Incident electrons may be temporarily trapped in the potential well formed by the potential of the target molecules in combination with the centrifugal potential of the incident electron, leading to the formation of a resonance (unstable anion with a short lifetime). This can happen from the ground state of the target (shape resonance) or involve one of its electronic excited states (Feshbach resonance). Resonances can be experimentally observed as local maxima in the total cross-section energy dependence [18] or through electron transmission measurements [26], which can be combined with a time of flight (TOF) analysis of the formed anion fragments (see ref. [26] for details). Although they are essentially inelastic processes, they normally appear as sharp peaks connected to the calculation of the integral elastic cross-sections [18,27]. The excess energy associated with the formation of the parent anion can be released along different pathways: autodetachment followed by energy relaxation, autodetachment followed by neutral dissociation or anionic dissociation. The latter channel is commonly known as dissociative electron attachment (DEA) and can be experimentally studied by means of a mass/charge analysis of the produced fragments after the electron attachment process. In most cases, this is carried out with a time of flight (TOF) spectrometer, which means that only the anionic fragments together with the parent anion can be analyzed. More complete experiments may incorporate a quadrupole mass spectrometer to analyze the neutral fragments; however, as far as we know, this information is not available for 1M5NI. 

In order to obtain a full description of the electron attachment (EA) process in terms of the EA cross-sections, we need to compile and critically discuss the results of integral elastic scattering cross-sections (accounting for inelastic channels in the calculation procedure), total electron scattering cross-section measurements and a charge/mass experimental analysis of the produced fragments. In addition, high-resolution electron transmission experiments are highly valuable to verify the positions and widths of the theoretically predicted resonances. We started our analysis with the energy dependence of our integral elastic cross-section that was calculated with the SMC method (see Section 4 for details). The results of this calculation are shown in Figure 3.

We considered that the local maxima in the IECS values shown in Figure 3 are due to resonant electron attachment, while the pure IECS follows a smooth energy dependence, as it corresponds to elastic processes. This assumption is supported by our TCS measurements, which confirm the positions of these resonances (see Figure 3). The electron attachment cross-sections were then derived by extracting the resonances from the elastic plus electron attachment cross-section curve. Some small resonances appearing at energies above 10 eV, which were not confirmed by any experimental evidence, were considered as pseudo-resonances, i.e., artifacts originated by failing to include the inelastic channels in the elastic scattering cross-section calculation. 

#### 2.3.2. Anion Yield Analysis

We investigated the anion fragmentation produced via dissociative electron attachment to 1M5NI by using a momentum imaging spectrometer, which has been described in detail previously [28]. The most relevant details of the experimental systems are described in Section 4.

Figure 4 shows the mass-resolved relative yields of anions produced in a dissociative electron attachment to 1M5NI in the resonant 3.1–4.7 eV energy range where the anion product yields were observed to be highest. The most abundant anions are NO_2_^−^ and CN^−^, followed by heavier anions having lost their neutral O, OH, CH_3_, NO and/or NO_2_ radicals. These results agree with those previously obtained by Tanzer et al. [29]. Note that the mass resolution (m/Δm ≈ 20) of the present momentum imaging spectrometer does not allow us to distinguish the number of hydrogen atoms remaining in most cases. At these attachment energies, there is no significant yield of H^−^ produced from the sample. For the 3.1 eV and 4.7 eV measurements, a small peak is clearly visible at m/q = 35 u, corresponding to either C_3_^−^ or H_2_O_2_^−^. In addition, clearly visible at all the energies is a peak around m/q = 54 u, which is possibly due to C_2_N_2_H_x_^−^ for x = 1, 2 or 3. 

The branching ratios (BRs), representing the contribution of a specific anion with respect to the sum of all the detected fragmentation channels, were measured for 1M5NI and are shown in Figure 5. At 3.1 eV electron energy, the contribution of NO_2_^−^ accounted for ~58% of the total anion yield, which reduced to ~35% at 4.7 eV. The relative cross-section for NO_2_^−^ production exhibited a peak around 3 eV and decreased as the electron energy increased [28], which is consistent with our observed reduction of the BR for NO_2_^−^. In contrast, the BRs for CN^−^ and CNO^−^ increased as the electron energy increased, reaching a maximum of 21% and 10%, respectively. The BRs of the other anions in Figure 5 (right panel) exhibited little variation in the energy range of 3.1–4.2 eV; they increased slightly between 4.2 and 4.7 eV. While none of these fragments contributed more than 10%, their combined yield accounts for 23–30% of the total anion yield.

The kinetic energy distributions for the five most abundant anions produced in DEA to 1M5NI are shown in Figure 6. The inset in each panel shows a central slice of each 3-dimensional momentum image. In all cases, the anion momentum is highly isotropic. The anion fragments’ kinetic energy distributions all peak near 0 eV, with little variation across the present range of incident electron energies. Perhaps most remarkable is the CN^−^ kinetic energy distribution, which is significantly broader than that of NO_2_^−^ and the heavier fragments. This suggests a three-body breakup or a stochastic mechanism in the dissociation of CN^−^, whereby the available energy is broadly distributed into any of the available nuclear degrees of freedom, in addition to the CN^−^ kinetic energy. Clearly, significant motion or rearrangement of the C_3_N_3_ ring, the NO_2_ or the CH_3_ moieties is required for the release of CN^−^. This contrasts with the dominant NO_2_^−^ dissociation, which may only require stretching of the nitro C–N bond. 

#### 2.3.3. Total Ionization Cross-Sections and Cation Formation by Electron Impact

Although electron impact ionization is probably the most relevant process in radiation damage and plasma processing applications, in general, related cross-section data are not abundant in the literature, especially for complex polyatomic molecules such as 1M5NI, which we are studying here. For the higher energies, the first Born approximation is commonly used to calculate the total ionization cross-sections by means of the Born–Bethe formula [30], while assuming an independent atom representation, which is a good approach for energies above 200–300 eV. The calculation procedures considering the distorted waves within the second Born approximation (DWSBA) allow us to extend this method to lower energies, which provide information on the single and double differential ionization cross-sections. In order to reproduce the maximum cross-sections around 60–90 eV, more elaborated calculations, such as the binary-encounter-Bethe (BEB) formulation [31], have been demonstrated to provide accurate data, within 10%, for a large number of molecular targets. A comparative study including the results for different approximations can be found in ref. [32]. Our screening-corrected additivity rule (IAM-SCAR), which considers an independent atom representation, corrects for the atomic screening within the molecule and includes relativistic and velocity-dependent effects [33], also competes with the BEB method. Both methods generally agree within their respective uncertainty limits for energies above 20–30 eV. For the lower energies, these approximations are not accurate enough and more reliable data would require employing more sophisticated ab initio methods, such as the R-matrix [34] and the convergent close coupling (CCC) approaches [35]. The limitation of these methods is mainly the size of the target to be treated, which is restricted mainly to atoms and small molecules. We have not found any experimental results on the total or partial ionization cross-sections in the literature. Itälä et al. [36] measured the photo fragmentation patterns of 1M4NI by soft X-ray synchrotron radiation. The photon decomposition of nitroimidazole compounds was investigated by Yu and Bernstein [37] and the dissociation of nitroimidazole ions was studied by Feketová et al. [38]. Since the amount of available data is not enough to build a reliable ionization data set, we followed a procedure based on our own calculation. We used our IAM-SCAR method to calculate the total electron impact ionization cross-sections from the threshold up to 1000 eV incident energy according to the procedure described in [33]. The corresponding results are given at the end of Section 2. 

Cation formation by electron impact can be experimentally determined by analyzing the mass spectrum and intensity of the different positive ions formed. Figure 7 shows the mass spectrum of the cationic species formed by the collision of 67 eV electrons with 1M5NI molecules as recorded with the time-of-flight spectrometer described in Section 4. From the mass analysis, cations can be assigned within the mass resolution limitation (~2 u). As this figure shows, the most intense positive ion formed corresponds to the parent ion (C_4_H_5_N_3_O_2_^+^) with 127 u. Other relevant features are found at 56 u, assigned to the C_2_H_4_N_2_^+^ (diazoethane) ion; 45 u, to the CH_3_NO^+^ (formaldoxine) ion; 29 u, to the CHO^+^ (Formyl radical) ion; and 15 u, assigned to the NH^+^ (imidogen radical) ion. The reactions producing these molecular fragments need to be investigated in order to evaluate the radiosensitizing properties of 1M5NI and its potential toxicity. Concerning the most representative fragments, diazoethane is a metabolite and its role has not been extensively studied. It has been characterized as highly reactive with specific sites of the O and N atoms of relevant biomolecules, such as uridine and thymidine [39]. Formaldoxime is highly reactive, since its carbonyl group (–C=O) undergoes different chemical reactions, such as reduction, oxidation and hydrolysis, thus acting as an efficient antibiotic, although it is limited by its toxicity. The formyl radical is very active in producing H^−^ anions by low-energy electron attachment [40], which, in turn, easily initiates reactions with sensitive biomolecules and imidogen; in combination with H_2_O, it initiates the prototypical amidation reaction of the O-H bonds [41].

**Figure 7 ijms-24-12182-f007:**
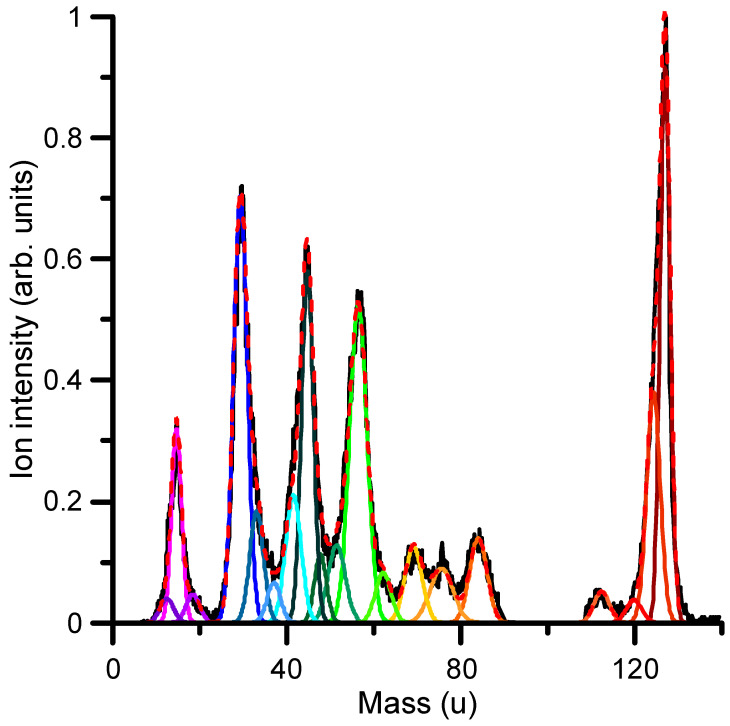
Mass analysis of the positive ions formed after the collision of 67 eV electrons with 1M5NI molecules (see the discussion in the text to identify cationic species that are shown in Table 2). Different color lines represent the Gaussian deconvolution of the spectrum.

#### 2.3.4. Differential Inelastic Cross-Sections

The relevant information to define the track structure of electrons passing through molecular media are the differential inelastic cross-sections. This information is provided by the double differential cross-sections (DDCS), which give the scattering angle distributions as a function of the energy transferred to the target molecule. Taking direct measurements of the DDCS for the whole impact energy range covered in this study, considering all the possible transferred energies, would be tedious and these measurements are generally not available for any molecular target. Calculations are more accessible but generally require drastic approximations to cover the wide impact and transferred energy ranges required for modelling purposes. One of the most commonly used approximations is based on the distorted wave second Born approximation (DWSBA) [42], which has been extensively used for representative molecules as water both in the gas [43] and condensed [44] phases. However, to our knowledge, no approximate calculations of the DDCS of complex molecules, such as 1M5NI, have been published. We thus followed the semiempirical procedure proposed for other molecular targets, such as pyridine [45]. From measurements of the angular distributions of scattered electrons performed for different incident energies (*E*), we proposed the following formula, which depends on the energy transferred to the medium (Δ*E*) and the corresponding differential elastic cross-sections: (1)$$\frac{d^{2}\sigma (E)}{d\Omega \Delta E}\propto {\left(\frac{d\sigma (E)}{d\Omega }\right)}_{el}^{1-\frac{k\Delta E}{E}},$$
where $\frac{d\sigma (E)}{d\Omega }$ represents the differential elastic cross section and *k* is a parameter to modify the weight of the dependence on Δ*E*/*E*. The value of *k* should be derived for each considered target molecule from direct measurements of the DDCS at the representative electron impact and transferred energies. For 1M5NI, we measured the angular distribution of scattered electrons corresponding to the generation of the parent ion and the three more intense peaks of the fragmentation mass spectrum (see Figure 8). In all cases, the incident and the transferred energies were found to be 97 and 30 eV, respectively. The results are shown in Figure 8. As can be seen in this figure, the observed angular distribution of the scattered electrons does not depend on the produced cation and presents a good agreement with that derived from Equation (1) for *k* = 1.3.

#### 2.3.5. Rotational, Vibrational and Electronic Excitation Cross-Sections

1M5NI is a polar molecule with a strong permanent dipole moment of 4.4 D; therefore, dipole rotational excitations by electron impact are relevant processes. However, the average rotational excitation transferred to the molecule at 300 K temperature is 0.617 meV, which is too low to be resolved by any of the experiments considered in this study. From a theoretical point of view, the calculation methods used in this study assume that the nuclei are fixed during the collision; hence, neither rotational nor vibrational excitation processes could be calculated. For the lower energies, in which we considered the SMC calculation for the elastic scattering to be accurate, an indirect way to account for the dipole interactions is to include the dipole moment in the scattering potential and apply the so-called Born correction [25] to the higher order partial waves. This implies an introduction of minimum transferred energy to avoid the 0-angle singularity of the Born approximation. The cross-section values obtained with this procedure are commonly termed “rotationally summed elastic cross-sections” to distinguish from the “pure” elastic cross-sections that are calculated without the Born correction (see [46] and references therein for details). This is very important when comparing theoretical and experimental elastic cross-sections (note that, as aforementioned, most of the experimental elastic cross-sections do not properly resolve rotational excitations).

Rotational excitation cross-sections can also be calculated independently by assuming the molecule as a rigid rotor and applying the first Born approximation (FBA). Although the incident energy could be very low, down to 0.1 eV in this case, the average rotational excitation mentioned above is at least three orders of magnitude lower and therefore may validate the FBA approach. Details on this calculation method are given in the case of pyridine by Sieradzka et al. [47]. The differential and integral cross-sections calculated with this procedure are shown in Figure 9a,b, respectively. Since they have been derived from a crude approximation, we recommend considering them only as a qualitative indication of the expected energy dependence, yet using their absolute values would require additional verifications (numerical results of the full calculation are given in Appendix A).

According to the self-consistent procedure we followed to obtain the recommended integral cross-sections, the total inelastic cross-section is derived by subtracting the pure integral elastic shown in Figure 10 from the reference TCS + MA values of Table 1. The electron attachment cross-sections are derived from the resonance analysis described above. After subtracting the total ionization cross-sections from the remaining inelastic channels, we separated the electronic and vibrational excitation cross-section by sampling their magnitudes from their respective thresholds to calculate the sum of their contributions at any energy, in order to be consistent with the reference TCS values at that energy. Finally, as previously mentioned, the rotational excitation cross-sections were estimated through an independent FBA calculation and are not included in the reference TCS values, i.e., they are not considered in this self-consistency procedure. The recommended cross-section data obtained with this method are shown in Table 3 and plotted in Figure 10 (the rotational excitations shown in Figure 8 are not considered in this figure). As shown in Table 3, the sum of the recommended integral cross-sections is in good agreement with the reference TCS data discussed in Section 2.1.

## 3. Discussion

The results presented in Section 2 constitute a complete dataset of self-consistent scattering cross-sections ready to be used for modelling electron transport in biologically relevant media, such as water containing traces of 1M5NI. This is a common situation in many radiobiological experiments [48] that are devoted to quantifying the living cell (assumed to be water) survival fractions with and without additional radiosensitizers. This data set is self-consistent in the sense that the sum of the recommended integral cross-sections of all the considered scattering channels (elastic, ionization, electronic excitation, vibrational excitation and electron attachment) at a given energy coincides with the reference value at that energy. These reference data were derived from our accurate (within 5%), previously measured, total electron scattering cross-sections together with the present calculation to correct the elastic “missing angle” effect (see above) and were complemented with our IAM-SCARI calculations up to 1000 eV impact energy. As 1M5NI is a polar molecule, additional electron scattering rotational excitation cross-sections were calculated by means of the Born approximation. Although this is a simple approximation, it elucidates an idea of the energy dependence of rotational excitation processes and can be used in applications in which heating processes may be relevant. For any of the scattering processes considered here, there are no previous theoretical or experimental cross-section data available in the literature. However, considering previous studies in which we applied a similar method to that proposed here to other molecules, such as water [49], pyridine [42] or benzene [50], we can estimate a 10% uncertainty limit to the most relevant scattering channels (elastic, ionization and electronic excitation) and up to 20–25% to the remaining channels (vibrational excitation and electron attachment). For the electron impact rotational excitations, we only provided qualitative information. 

Since the damaging effects of molecular radiosensitizers are connected to the production of radicals by electron-induced molecular dissociation, we have studied the generation of anionic fragments by low-energy electrons and the formation of cationic fragments by relatively high-energy electrons. The observed anionic and cationic fragmentation patterns agree with the prediction of previous studies [14,15,18,29,36,37] and confirm the potential activity of 1M5NI as a radiosensitizer. These are the most accurate data we have been able to derive by combining our available theoretical and experimental tools. This level of accuracy could be improved by designing new experiments and/or calculations to determine the absolute values of the anion, cation and neutral fragment production yields. The electronic, vibrational and rotational excitation cross-sections would also need further verification. 

Using the present data set, electron track simulations can be performed in order to obtain information about the number and type of dissociative processes induced to 1M5NI molecules by an electron beam. As far as we know, radiobiological experiments correlating the electron-induced radical generation in molecular radiosensitizers with the observed molecular damage are not reported in the literature. However, we are planning to repeat experiments such as those reported in ref. [48] by adding 1M5NI to the living cell target in order to confirm and quantify its radiosensitizing effect.

## 4. Materials and Methods

### 4.1. Calculation Methods

For the lower electron impact energies (<15 eV), the elastic differential and integral cross-sections were computed with the SMC method [23]. The calculations relied on the fixed-nuclei approximation, where the geometry was obtained with the density functional theory using the B3LYP functional and the aug-cc-pVDZ basis set. The restricted Hartree–Fock approximation was employed to describe the electronic ground state with the Cartesian Gaussian functions given in ref. [24]. As mentioned before, the electronic excitation channels were not considered in the scattering calculations. Besides the usual spin-adapted configurations associated with the static-exchange approximation, we further accounted for the polarization effects by introducing a set of spin-adapted configurations built from single excitations. For this purpose, we used the orbital energy criterion introduced in refs. [51,52], with an energy cutoff of ε_cut_ = 1.43 Hartree. Additionally, the canonical virtual orbitals were replaced by modified virtual orbitals generated in the field of the cation of charge +8. To avoid possible numerical problems, the singular value decomposition technique was utilized. We removed the combination of configurations associated with the three lowest singular values for the A′ symmetry and with the lowest singular value for the A″ symmetry. The long-range electron–molecule dipolar interaction was accounted for in the calculations via the Born-closure procedure [53]. Additional details about the calculations can be found in a previous publication (see supplementary information in ref. [18]) with some technical details available in refs. [54,55,56,57]. 

For the higher collision energies (>15 eV), we employed the IAM-SCAR+I method to obtain differential and integral elastic cross-sections, as well as integral inelastic electronic excitation and ionization cross-sections. This method has been described in detail [20] and its reliability has been thoroughly verified [58,59,60]. Briefly, the molecular target is considered as an aggregate of its individual atoms. Each atom is represented by an “ab initio” optical potential, where the real part accounts for the elastic scattering, while the imaginary part represents the inelastic processes, considered as the “absorption part”. The differential scattering cross-sections (DCSs) were obtained from the atomic data by the screening-corrected additivity rule (SCAR) procedure, incorporating interference (I) corrections by summing all the atomic amplitudes, where the phase coefficients are included. Then, by integrating over all the scattered angular range, the integral scattering cross-sections (ICSs) were obtained. Finally, the rotational excitation cross-sections stem from the first-Born approximation.

### 4.2. Experimental Methods

The total electron scattering cross-sections, which have been used as reference values to evaluate the self-consistency of the present data set, were taken from our recent measurements [18] that were performed with a magnetically confined electron transmission apparatus [19]. Details on this experimental setup can be found in ref. [19].

An anion fragment momentum imaging spectrometer was employed at Lawrence Berkeley National Laboratory to analyze the relative yields and kinetic energies of the anion fragments produced by the dissociative electron attachment (DEA) to 1M5NI. The experimental arrangement to perform these measurements has been previously described in detail [28], thus we include here only the information most relevant for the present experiments. Briefly, a stainless-steel capillary was employed to produce an effusive jet of molecules, which crosses orthogonally with a pulsed electron beam in a coaxial magnetic field inside the spectrometer. At one end of a gas manifold system, a glass sample holder containing approximately 10 g of 1M5NI was heated to a temperature range of 40–50 °C. The gas manifold feeding the gas jet capillary was also gradually heated by increasing the temperature (<80 °C). This process caused the sublimated 1M5NI vapor to accumulate in the gas manifold and build up a pressure of 10–100 mTorr, before the gas was introduced into the sample gas inlet and capillary. The electron beam energy spread (0.5 eV full width at half-maximum) and the absolute electron beam energy was determined and verified before and after the present experiments by measuring the anion yields across the thermodynamic threshold for O^−^ production from CO_2_. The anion fragment momentum was calibrated against the well-known O^−^ momentum distribution from DEA to O_2_. The time-of-flight and positions of each ion hit were recorded by a time- and position-sensitive multichannel plate detector with delay-line readout in an event list-mode format. The raw position and time data were sorted and converted to momenta after each experiment was completed.

The induced cationic fragmentation and the double differential ionization cross-sections were measured with a reaction microscope (ReMi) multi-particle coincidence spectrometer [61,62]. The schematic diagram of the apparatus is shown in Figure 11.

A pulsed electron beam was used as projectile beam. It was produced with a photoemission source (electron gun) containing a tantalum photocathode, which was illuminated with UV light (266 nm) of 0.5 ns pulse duration from a diode laser with a 40 kHz repetition rate. The electron gun is mounted within the drift tube of the spectrometer. The produced electron beam propagates along the spectrometer axis (*Z* axis) and is dumped into a hole of 8 mm radius on the electron detector. Electrostatic lenses in the electron gun assembly and an axial magnetic guiding field focus the beam to a diameter of ~0.5 mm at the interaction region. The 1-Methyl-5-Nitroimidazole (1Me5NI) target was prepared using a syringe setup, as shown in Figure 12. This newly built gas target was required since the standard supersonic gas target cannot be employed due to the sample’s particularly low vapor pressure of 0.003 mbar at room temperature. 1Me5NI powder was loaded inside a syringe, which was fixed on an xyz-manipulator. The target sample underwent sublimation and the vapor was introduced to the interaction region within the spectrometer by moving the setup along the X-direction in the lab frame, such that the tip of the needle of the syringe was close to, but not hitting the electron beam. To increase the target density, the sample was slightly heated up to 56 °C. For guiding of the outgoing electrons to the electron detector, homogenous electric and magnetic fields of 4 V/cm and 7.4 G, respectively, were used. For this purpose, the syringe needle was kept on the proper spectrometer potential. After the electrons reached the detector, the electric spectrometer field was ramped up to 22.6 V/cm for extraction of the ions. In order to prevent the low-momentum ions (e.g., parent ions) from colliding with the electron gun assembly, a pulsed voltage higher than the voltage of the spectrometer ring at the interaction region was applied to the syringe needle. In this way, the low-momentum ionic fragments were pushed transversally out of the interaction region and subsequently by-passed the electron gun and reached the ion detector. Both the electron and the ion detectors are time- and position-sensitive microchannel plate detectors with hexagonal (multi-hit) delay line readout. 

From the time and position information, the momenta and, therefore, the kinetic energies and the emission angles of the detected electrons were reconstructed. The acceptance angle was close to 4π for electrons with energies between 0.3 eV and 18 eV. Electrons moving in small forward or backward angles and those with energies below 0.3 eV were not detected due to the presence of the central hole in the electron detector. The solid angle coverage for the intact parent ions was almost 100 deg. However, the initial momentum information of the ionic fragments was lost due the inhomogeneous electric field used to push the fragments away from the center. Therefore, only the time-of-flight information of the ions was used to differentiate the different fragmentation channels by the ionic fragment mass.

## Figures and Tables

**Figure 1 ijms-24-12182-f001:**
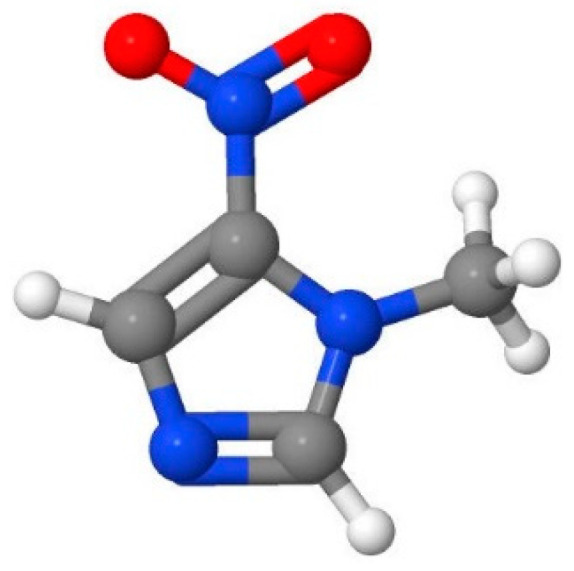
Molecule structure of 1-Methyl-5-Nitroimidazole (1M5NI).

**Figure 2 ijms-24-12182-f002:**
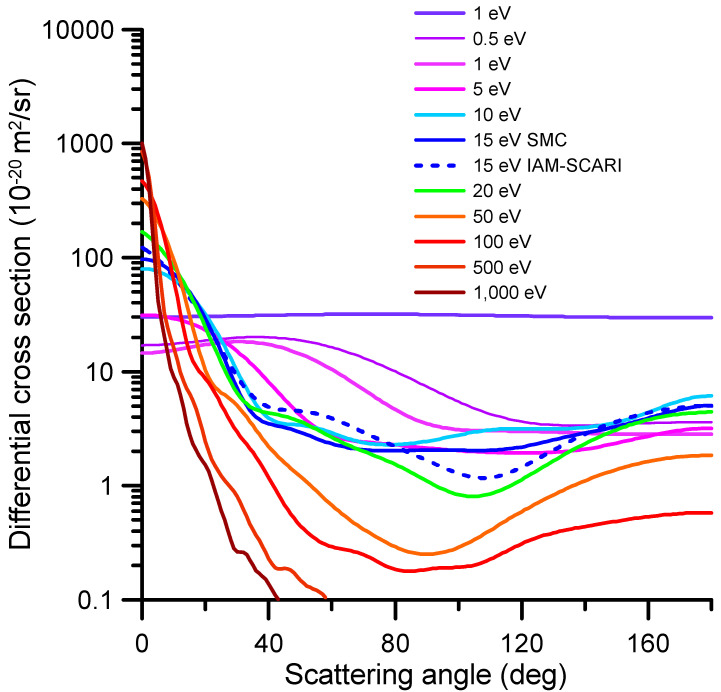
Representative differential elastic cross-section calculated with the SMC method for the 0.1–15 eV impact energy range and the IAM-SCARI from 15 to 1000 eV. The agreement between both methods at 15 eV is discussed in the text (numerical results of the full calculation are given in Appendix A).

**Figure 3 ijms-24-12182-f003:**
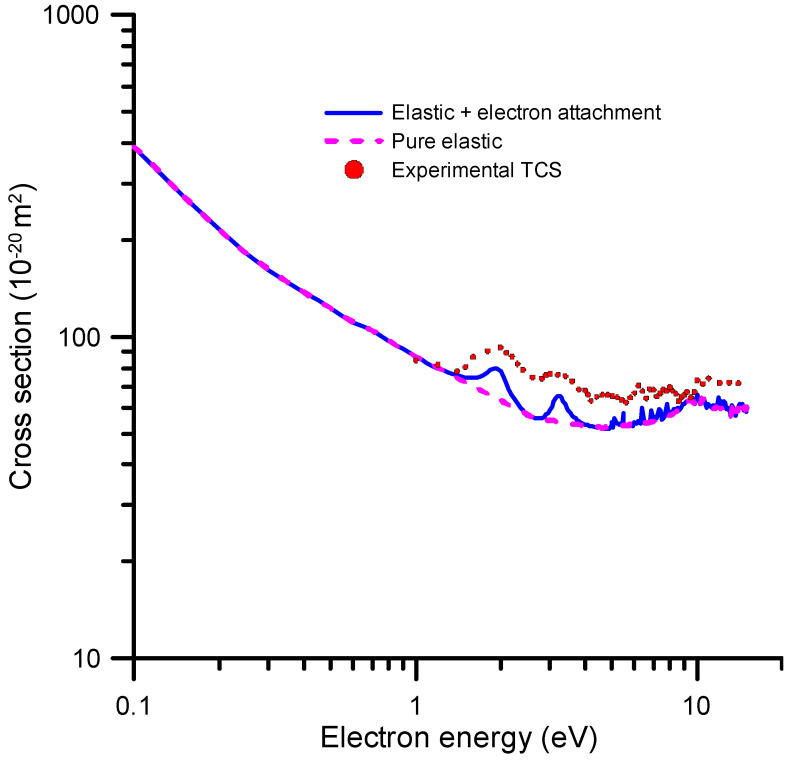
Integral elastic cross-section calculated with the SMC method (see text for details).

**Figure 4 ijms-24-12182-f004:**
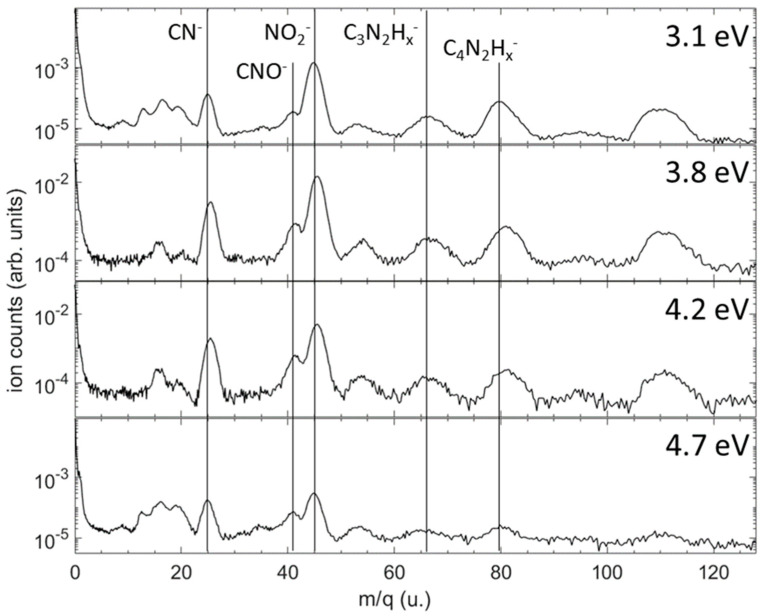
Time of flight mass spectra showing the relative yields of anion fragments produced in dissociative electron attachment to 1M5NI at four incident energies. The vertical scale is not normalized between the four electron energies; it shows the number of ions on a logarithmic scale for each measurement at a fixed electron beam energy. Vertical lines indicate the most prominent anion fragments. The region around 16 a.m.u is subject to contamination from DEA to H_2_O, O_2_ and CO_2_, all of which produce O^−^ following attachment of electrons on the high-energy side of the electron beam energy distribution.

**Figure 5 ijms-24-12182-f005:**
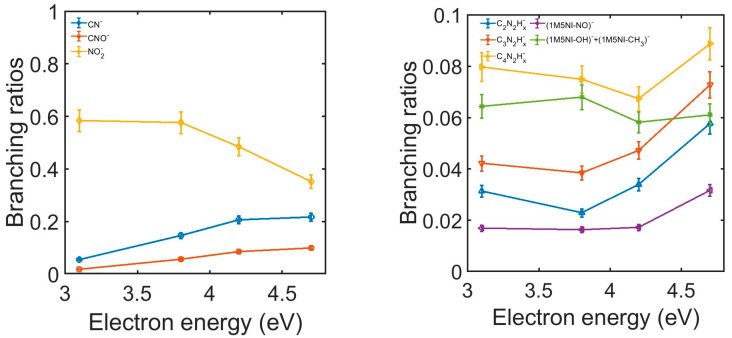
The experimental 1M5NI branching ratios of the negative ions formed as a function of electron energy. Left panel: ions CN^−^ (blue), CNO^−^ (orange) and NO_2_^−^ (yellow). Right panel: ions C_2_N_2_H_x_^−^ (blue), C_3_N_2_H_x_^−^ (orange), C_4_N_2_Hx^−^ (yellow), NO loss (violet) and the sum of OH and CH_3_ loss (green).

**Figure 6 ijms-24-12182-f006:**
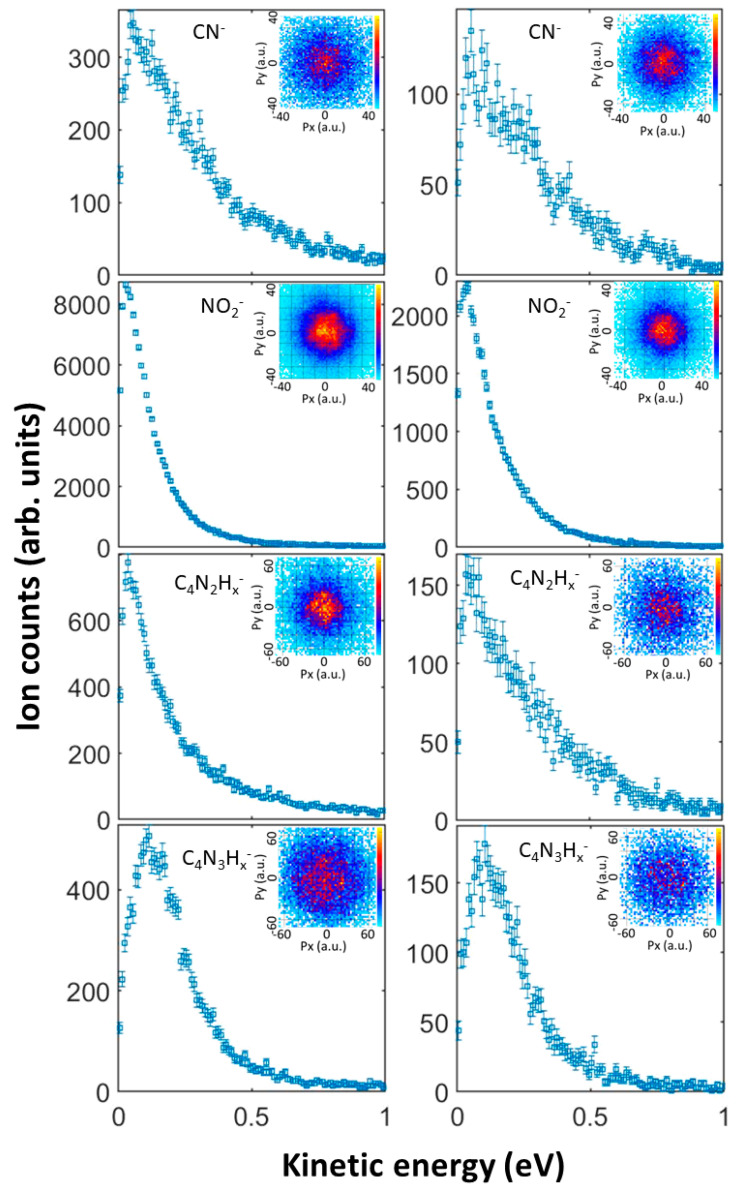
Measured mass-resolved kinetic energy distributions of the anion fragments, and momentum-sliced images (insets) in atomic units (a.u.) for anion fragments produced in dissociative attachment of 3.1 eV (left column) and 4.2 eV (right column) electrons to 1M5NI. The incident electron is in the +Py direction in the momentum images.

**Figure 8 ijms-24-12182-f008:**
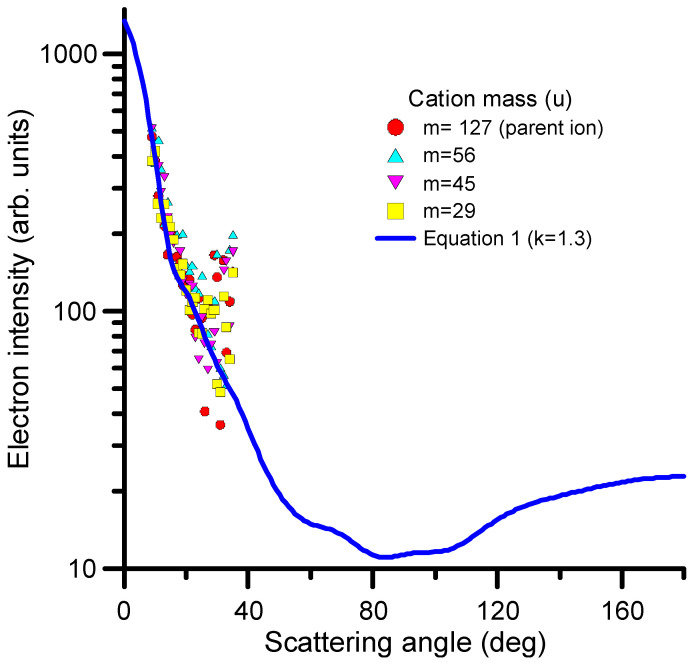
Angular distribution of scattered electrons after the collision of 97 eV electrons with 1M5NI transferring 30 eV to the target to produce cationic species (see figure legend).

**Figure 9 ijms-24-12182-f009:**
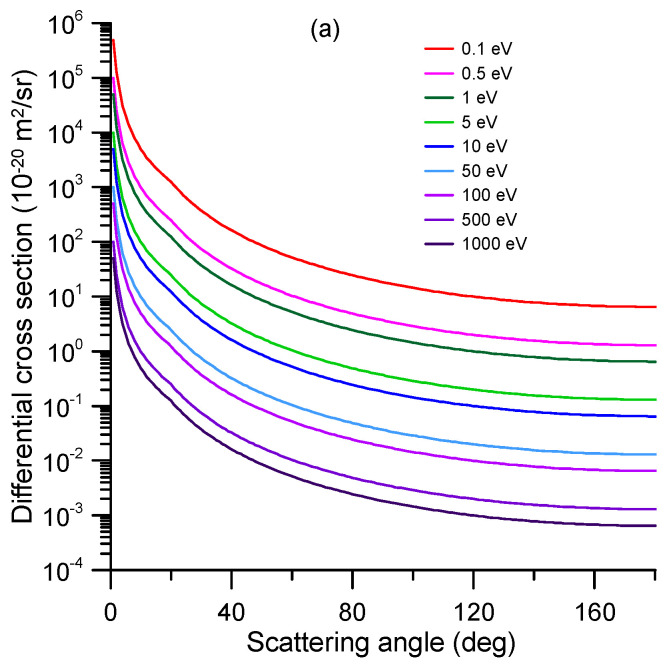
Rotational excitation cross-sections of 1M5NI calculated with the first Born approximation. (**a**) Differential cross-sections; (**b**) integral cross-sections.

**Figure 10 ijms-24-12182-f010:**
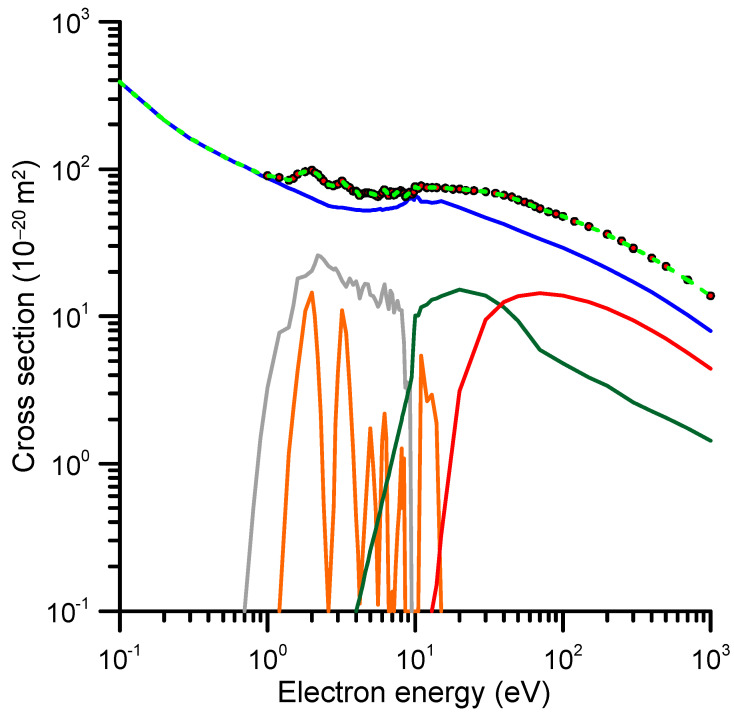
Recommended integral cross-sections (CS). **—**, elastic CS; **—**, ionization CS; **—**, electronic excitation CS; **—**, vibrational excitation CS; **—**, electron attachment CS; **---**, sum of all the considered scattering channels of 1M5NI (excluding the rotational excitation; see text for details); ●, TCS + MA reference data (experimental values corrected for elastic missing angles; see text).

**Figure 11 ijms-24-12182-f011:**
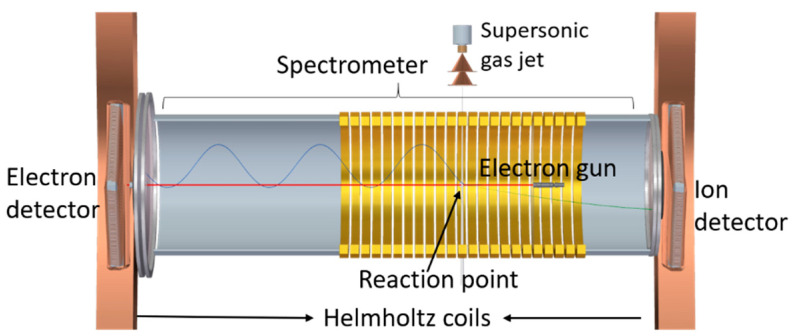
Schematic view of the reaction microscope (ReMi) multi-particle coincidence spectrometer used to measure the induced cationic fragmentation and the double differential ionization cross-sections.

**Figure 12 ijms-24-12182-f012:**
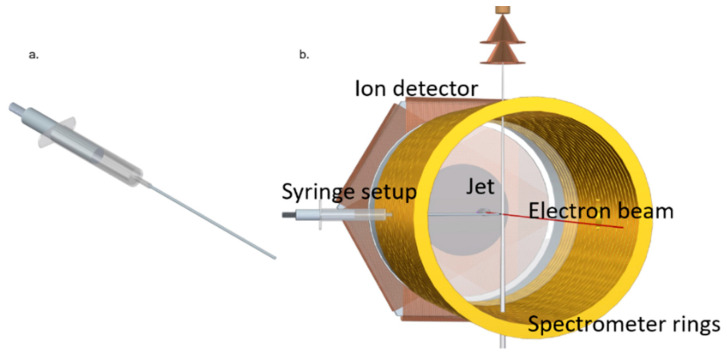
(**a**) The target vapor source consists of a standard medical syringe with a 1 mm inner diameter, 10 cm long stainless-steel needle. (**b**) View into the electrode array of the reaction microscope with the syringe needle introduced horizontally through a hole in a spectrometer ring electrode. In the back, the ion detector is visible, while the spectrometer drift tube and the electron detector on the front side are not shown. The vertically aligned helium supersonic gas jet was used in the present measurements for calibration of the spectrometer fields.

**Table 1 ijms-24-12182-t001:** Experimental results of the total electron scattering by 1M5NI cross-sections and the corresponding corrected values, accounting for the elastic scattering in the “missing angles (MA)” (see text for details).

E (eV)	TCS_exp (10^−20^ m^2^)	TCS + MA (10^−20^ m^2^)	E (eV)	TCS_exp (10^−20^ m^2^)	TCS + MA (10^−20^ m^2^)
1	84.4	90.13	7.8	67.8	69.47
1.2	81.8	87.77	8	69.7	71.47
1.4	78	84.10	8.2	67.6	69.46
1.5	80.4	86.52	8.4	66.4	68.34
1.6	86.3	92.40	8.6	62.9	64.93
1.8	90.2	96.13	8.8	64.8	66.92
2	92.6	98.16	9	65.7	67.93
2.1	89.2	94.49	9.2	67.3	69.66
2.2	86.8	91.77	9.5	64.7	67.26
2.3	82.3	86.93	9.8	67.2	69.93
2.4	78.8	83.06	10	73.2	75.99
2.6	74.7	78.23	10.5	71.4	74.13
2.8	74	76.85	11	74.2	76.86
2.9	75.7	78.27	12	71.8	74.77
3	76.7	79.06	13	72	74.90
3.2	76.1	78.20	14	71.6	74.84
3.4	75.5	77.57	16	71.5	74.12
3.6	72.8	74.97	18	70.5	73.05
3.8	68.5	70.88	20	70.7	73.18
4	68.1	70.72	22	69.1	71.52
4.1	66	68.72	25	69	71.34
4.2	63.2	66.02	30	67.9	70.13
4.4	65	67.98	35	66.4	68.55
4.6	65.9	69.00	40	64.3	66.37
4.8	65.9	69.06	45	62.7	64.71
5	65.3	68.46	50	59.7	61.65
5.2	64.6	67.68	55	57.9	59.80
5.4	63.5	66.46	60	56.3	58.16
5.6	62.4	65.17	65	53.9	55.72
5.8	64.5	67.07	70	52.1	53.88
6	66.9	69.24	80	49.6	51.32
6.2	70.4	72.51	90	47.9	49.56
6.4	67.6	69.51	100	46.2	47.81
6.6	64.4	66.12	120	42.6	44.13
6.8	68.6	70.18	150	39.3	40.74
7	68.3	69.80	200	34.8	36.12
7.2	65.1	66.57	250	31.3	32.54
7.4	66.8	68.31	300	27.6	28.78
7.6	68.4	69.98			

**Table 2 ijms-24-12182-t002:** Identification of the main cationic species formed after the collision of 67 eV electrons with 1M5NI molecules (see also Figure 7).

Mass (u)	Chemical Composition	Relative Intensity
127	C_4_H_5_N_3_O_2_^+^ (1M5NI ion)	1
124	C_4_H_2_N_3_O_2_^+^ (1M5NI-3H)	0.415
112	C_4_H_4_N_2_O_2_^+^ (1M5NI-NH)	0.0564
84	C_4_HNO (Acrylonitrile-ketone)	0.152
75	C_2_H_5_NO_2_ (Glycine)	0.0978
69	C_2_H_3_N_3_ (Triazole)	0.137
62	CH_4_NO (Hydroxymethyl-amino-oxy radical)	0.0900
56	C_2_H_4_N_2_ (Diazoethane)	0.571
51	C_3_HN (Cyanoacetylene)	0.139
48	H_2_NO_2_^+^ (Nitronium)	0.124
45	CH_3_NO^+^ (Formaldoxime)	0.633
42	C_2_OH_2_^+^ (Ketene)	0.230
38	C_2_N (Carbon Cyanide radical)	0.0720
33	H_3_NO (Hydroxylamine)	0.201
29	CHO (Formyl radical)	0.768
18	H_2_O^+^ (Water)	0.0512
15	NH (Imidogen)	0.349
12	C (Carbon)	0.0435

**Table 3 ijms-24-12182-t003:** Recommended integral cross-sections for electron scattering cross-sections from 1M5NI, in units of 10^−20^ m^2^, derived from the proposed self-consistent procedure (see text for details).

Energy (eV)	Elastic	Electron Attachment	Vibrational Excitation	Electronic Excitation	Ionization	SUM
0.1	390					390
0.2	214					214
0.3	162					162
0.4	138					138
0.5	122					122
0.7	104		0.1			104
1.0	86.8		3.28			90.1
1.5	72.2	2.39	11.9			86.5
2.0	63.3	14.5	20.3			98.2
3.0	55.0	4.12	20.0			79.1
4.0	52.9	0.425	17.3	0.1		70.7
5.0	52.5	1.74	13.9	0.257		68.5
7.0	54.4	0.136	14.2	1.07		69.8
10	65.8	0.1		10.1		76.0
15	60.5	0.1		13.7	0.328	74.5
20	54.9			15.2	3.11	73.2
30	46.8			13.8	9.52	70.1
40	42.3			11.5	12.6	66.4
50	38.6			9.29	13.7	61.6
70	33.6			11.5	12.5	66.4
100	29.1			4.78	13.9	47.8
150	24.3			3.83	12.6	40.7
200	21.2			3.38	11.3	35.9
300	17.1			2.62	9.41	29.2
400	14.6			2.27	8.04	24.89
500	12.7			2.04	7.06	21.8
700	10.2			1.74	5.68	17.7
1000	7.95			1.43	4.42	13.8

## Data Availability

Not applicable.

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
