# Peer review of "Electron Scattering from 1-Methyl-5-Nitroimidazole: Cross-Sections for Modeling Electron Transport through Potential Radiosensitizers"

_ijms, 2023, doi:10.3390/ijms241512182_

Round 1

Reviewer 1 Report

This paper focuses on electron scattering cross sections from 1-Methyl-5-Nitroimidazole (1M5NI) molecules, which is relevant to evaluate the potential role of 1M5NI as a molecular radiosensitizer. The authors have used two different schemes, the Schwinger Multichannel (SMC) method for lower energies and the Independent Atom Model-based Screening Corrected Additivity Rule with Interferences (IAM-SCARI) for higher energies, to calculate elastic scattering cross sections. The total ionization cross sections have also been calculated using the IAM-SCARI method and complemented with experimental values of the induced cationic fragmentation by electron impact. The authors have also presented measurements of double differential ionization cross sections, anion fragment yields, and kinetic energies by dissociative electron attachment using a reaction microscope multi-particle coincidence spectrometer and a momentum imaging spectrometer. The cross sections for other inelastic channels have been derived using a self-consistent procedure by sampling their values at a given energy.

Abstract

The abstract provides a comprehensive overview of the methods and techniques used in the study and the results obtained. However, it does not clearly state the key findings of the study, which makes it difficult for readers to gauge the significance of the research. Also, it does not provide any information on the limitations of the study or potential implications of the results.

Introduction

The introduction is well-structured and provides a clear background and rationale for the study. However, the introduction could benefit from a more detailed explanation of the specific molecular properties and potential radiosensitizing mechanisms of 1-methyl-5-nitroimidazole (1M5NI), which is the focus of the study. Additionally, the introduction does not clearly state the research questions or hypotheses being addressed in the study, which could make it difficult for readers to understand the purpose and scope of the research.

Result

1.     Many of the calculated cross sections use approximations like the Born approximation, independent atom model, and distorted wave methods. These approximations introduce uncertainty in the absolute values.

2.     There are no calculations of double differential cross sections for inelastic scattering processes. The authors make an assumption based on analogous calculations for other molecules.

3.     The rotational excitation cross sections are calculated using a crude first-Born approximation. The authors themselves recommend treating these values only qualitatively.

Discussion

1.     The authors acknowledge that there is little or no experimental data available on many of the scattering cross sections, especially for inelastic channels. They have relied heavily on calculations and approximations. This introduces significant uncertainties in the absolute values of the cross sections.

2.     The anion and cation yields provide only qualitative insights into the dissociation of 1M5NI. More quantitative data on the fragmentation mechanisms and kinetics is lacking.

3.     While the authors attempt to estimate the uncertainty in the cross sections, they admit that the estimates for some channels may be quite high (up to 25%). More experimental validation is needed to improve the accuracy.

Author Response

Response to Reviewer 1

Reviewer:

This paper focuses on electron scattering cross sections from 1-Methyl-5-Nitroimidazole (1M5NI) molecules, which is relevant to evaluate the potential role of 1M5NI as a molecular radiosensitizer. The authors have used two different schemes, the Schwinger Multichannel (SMC) method for lower energies and the Independent Atom Model-based Screening Corrected Additivity Rule with Interferences (IAM-SCARI) for higher energies, to calculate elastic scattering cross sections. The total ionization cross sections have also been calculated using the IAM-SCARI method and complemented with experimental values of the induced cationic fragmentation by electron impact. The authors have also presented measurements of double differential ionization cross sections, anion fragment yields, and kinetic energies by dissociative electron attachment using a reaction microscope multi-particle coincidence spectrometer and a momentum imaging spectrometer. The cross sections for other inelastic channels have been derived using a self-consistent procedure by sampling their values at a given energy.

Authors’ reply:

We acknowledge his/her careful review of the paper and the general positive evaluation.

Reviewer:

Abstract

The abstract provides a comprehensive overview of the methods and techniques used in the study and the results obtained. However, it does not clearly state the key findings of the study, which makes it difficult for readers to gauge the significance of the research. Also, it does not provide any information on the limitations of the study or potential implications of the results

Authors’ reply:

Following the reviewer suggestions we have included in the abstract the following sentences:

“This cross-section data set is ready to be used for modelling electron induced radiation damage, at the molecular level, to biologically relevant media containing 1M5NI as potential radio-sensitizer. Nonetheless a proper evaluation of its radiosensityzing effects would require further radiobiological experiments.”

Which clearly state the significance of the present results, giving also an indication of their limitations to evaluate the biological induced damage.

Reviewer:

Introduction

The introduction is well-structured and provides a clear background and rationale for the study. However, the introduction could benefit from a more detailed explanation of the specific molecular properties and potential radiosensitizing mechanisms of 1-methyl-5-nitroimidazole (1M5NI), which is the focus of the study. Additionally, the introduction does not clearly state the research questions or hypotheses being addressed in the study, which could make it difficult for readers to understand the purpose and scope of the research.

Authors’ reply:

New sentences explaining the specific properties and potential radiosensitizing properties of 1M5NI, supported by an additional reference (Ref. 16), have been included in the introduction section. The list of references has been accordingly renumbered. The hypothesis and motivation of the present study has also been stated in the same paragraph.

Reviewer:

Result

  1. Many of the calculated cross sections use approximations like the Born approximation, independent atom model, and distorted wave methods. These approximations introduce uncertainty in the absolute values

Authors’ reply:

Within the self-consistent cross section data set (excluding the rotational excitation) provided in this study no calculations based on the Born approximation are included. As explained in section 2.2.,  the elastic cross sections have been calculated with our IAM-SCARI method which is based on the independent atom model but includes relevant improvement to make it valid, within 10% (as demonstrated in previous studies cited in the text), for impact energies above 15 eV. Below 15 eV we performed an “ab initio” calculation based on the Schwinger Multichannel Method (details of this method are given in Section 4). The overall uncertainty limit  assigned to these elastic cross sections is about 10%. As explained in section 2.3.3., the total ionization cross is also calculated with our IAM-SCARI procedure. As demonstrated in previous studies cited in the text this calculation is accurate enough (within 10%) for energies around and above that of the maximum ionization cross section (50 eV in this case). From the ionization threshold to this energy the present data are not reliable enough. However, as no accurate measurement have been found in the literature we have adopted these calculated values and recommended further experimental studies on near threshold ionization cross sections of 1M5NI.

  1. There are no calculations of double differential cross sections for inelastic scattering processes. The authors make an assumption based on analogous calculations for other molecules.

Authors’ reply:

As explained in section 2.3.4 accurate double differential cross sections (differential with respect to the energy transferred to the target and the scattering angle) are only available for atoms and small molecules. For more complicated molecules we proposed a semiempirical method which was proved to be accurate for relatively complicated molecules such as THF and pyrimidine. The semiempirical formula is represented by equation (1) and, as shown in Figure 8,  it is experimentally supported by our experimental double differential ionization cross sections as measured with the reaction microscope described in Section 4.  As far as we know, this is the most accurate way to introduce a full inelastic angular distribution function for modelling electron transport purposes.

  1. The rotational excitation cross sections are calculated using a crude first-Born approximation. The authors themselves recommend treating these values only qualitatively.

Authors’ reply:

Even the SMC “ab initio” method used in this study, as well as other sophisticated procedures such as the R-matrix or the Convergent Close Coupling, assume that nuclei are fixed during the collision process.  This approach is limiting the possibility of using them to calculate rotational excitation cross sections. For this reason the First-Born approximation (FBA) is commonly used to calculate these cross sections. We recognize this could be a crude approximation but, as explained in the text, the average energy transferred via rotational excitations to these molecules is of the order of 0.00001 eV and therefore is almost impossible to be experimentally verified. In these conditions, as explained in section 2.3.5., we consider the results of our FBA calculation are just qualitative, providing the expected energy dependence but not their absolute values. In most biomedical applications rotational excitation processes are not relevant and can be excluded from the modelling procedure. In fact, we have not included them in the present self-consistent procedure. However for those applications in which rotational excitations could be important (critical temperature dependence, for example) additional verifications would be needed. 

Discussion

  1. The authors acknowledge that there is little or no experimental data available on many of the scattering cross sections, especially for inelastic channels. They have relied heavily on calculations and approximations. This introduces significant uncertainties in the absolute values of the cross sections.

To derive the cross section data set presented in this study we have combined our available theoretical and experimental techniques with their respective accuracies. As  explained in section 2.1., the procedure followed in this study rely on our accurate total electron scattering  cross section (TCS) measurements with assigned uncertainties within 5%. These are the reference values and for any impact energy the sum of the cross section of all the scattering channels, with their respective uncertainties,  which are open at that energy should equal the TCS reference value. The most relevant scattering channels (elastic, ionization and electronic excitation) are derived from verified calculation methods which allow us to assign typical uncertainty limits of about 10 %. The uncertainty assigned to other less relevant processes (in terms of their contribution to the TCS)  can be higher in magnitude (15-25%) but the self-consistency of the proposed method ensures that their incidence in the application outputs won’t be significant. Nonetheless, the present dataset is open to incorporate more accurate data that could be  provided by new experiments or calculations. This consideration is now commented in the last paragraph of the Discussion section.

  1. The anion and cation yields provide only qualitative insights into the dissociation of 1M5NI. More quantitative data on the fragmentation mechanisms and kinetics is lacking.

As mentioned above, our present experimental techniques are not able to provide the absolute value of the induced anionic and cationic fragmentation. From our total electron attachment and total ionization cross section, in combination with the observed branching ratio, they could be derived but as stated now in the last paragraph of the Discussion section: “These are the most accurate data we have been able to derive by combining our available theoretical and experimental tools. This level of accuracy could be improved by designing new experiments and/or calculation to determine the absolute values of anion, cation and neutral fragment production yields. The electronic, vibrational and rotational excitation cross sections would also need further verifications.”

  1. While the authors attempt to estimate the uncertainty in the cross sections, they admit that the estimates for some channels may be quite high (up to 25%). More experimental validation is needed to improve the accuracy.

As mentioned above and explained in the text, the inelastic channels with assigned uncertainties within 15-25% mainly correspond to vibrational excitation and neutral dissociation processes and their contribution to the TCS value is less relevant than that of the main scattering channels (elastic, ionization and electronic excitation). Nonetheless we agree with the referee that “More experimental validation is needed to improve the accuracy.” This will be the subject of further investigations within our research group but it will require new experimental designs.

Reviewer 2 Report

Lozano, et al. present the manuscript titled "Electron scattering from 1-Methyl-5-Nitroimidazole: cross sections for modelling electron transport through potential radiosensitizers."  This manuscript provides an in-depth look at the electron scattering properties of 1M5N1 in a robust manner.  These results provide useful insights into their potential utilization in radiobiology.  

Few minor comments: 

1. Methods are well described with appropriate schematic diagrams to help the average reader, however, they should be moved to the front of the article to help the audience interpret and understand the results.  

2. The authors should provide more background information on 1M5N1 including the structural schematic to help the audience. 

3. The radiobiological implications of these results as well as any previous studies testing the use of 1M5N1 should be discussed in more detail than is provided in the discussion.

Author Response

Response to Reviewer 2

We acknowledge the positive evaluation of our work reported by the reviewer. All his/her minor comments have been addressed as follows:

Reviewer:

  1. Methods are well described with appropriate schematic diagrams to help the average reader, however, they should be moved to the front of the article to help the audience interpret and understand the results.

Authors’ reply:

We agree with the referee but the manuscript has been prepared according to the template provided by the editors in which the Methods section is placed on the forth position.

Reviewer:

  1. The authors should provide more background information on 1M5N1 including the structural schematic to help the audience

Authors’ reply:

A brief description of the radiosensitizing properties of 1M5NI have been included in the Introduction section and its molecular structure is now shown in the Figure 1 of the amended version of the paper.

Reviewer:

  1. The radiobiological implications of these results as well as any previous studies testing the use of 1M5N1 should be discussed in more detail than is provided in the discussion.

Authors’ reply:

As far as we know, radiobiological experiments including 1M5NI as radiosensitizers are not reported in the literature. However, in order to give more information about radiobiological implications of the present results, the last paragraph of Section 3 now reads as follows:

“Since damaging effects of molecular radiosensitizers are connected to the production of radicals by electron induced molecular dissociation, we have studied the generation of anionic fragments by low energy electrons and the formation of cationic fragments by relatively high energy electrons. The observed anionic and cationic fragmentation patterns agree with the prediction of previous studies [14, 15, 18, 29, 36, 37] and confirm the potential activity of 1M5NI as a radiosensitizer. These are the most accurate data we have been able to derive by combining our available theoretical and experimental tools. This level of accuracy could be improved by designing new experiments and/or calculation to deter-mine the absolute values of anion, cation and neutral fragment production yields. The electronic, vibrational and rotational excitation cross sections would also need further verifications.

Using the present data set, electron track simulations can be performed in order to obtain information about the number and type of dissociative processes induced to 1M5NI molecules by an electron beam. As far as we know, radiobiological experiments correlating the electron induced radical generation in molecular radiosensitizers with the observed molecular damage are not reported in the literature. However, we are planning to repeat experiments like that reported in Ref. [49] by adding 1M5NI to the living cell target in order to confirm and quantify its radiosensitizing effect.
